# The Academic Landscapes of Manufacturing Enterprise Performance and Environmental Sustainability: A Study of Commonalities and Differences

**DOI:** 10.3390/ijerph18073370

**Published:** 2021-03-24

**Authors:** Cristian Mejia, Yuya Kajikawa

**Affiliations:** 1Graduate School of Environment and Society, Tokyo Institute of Technology, Tokyo 108-0023, Japan; kajikawa@mot.titech.ac.jp; 2Institute for Future Initiatives, The University of Tokyo, Tokyo 113-0033, Japan

**Keywords:** manufacturing enterprise, performance, environmental sustainability, literature-based discovery, citation networks, natural language processing, manufacturing, bibliometrics

## Abstract

This article reviews literature on manufacturing enterprise performance (MEP) and environmental sustainability (ES) to identify their commonalities and distinguishing factors; it is expected to help determine gaps and paths for future research. Topics are classified based on patterns in the citation networks of 7308 and 6275 MEP and ES articles, respectively. Additionally, a semantic linkage was computed to reveal overlap in vocabulary between the two topics. A total of 17 and 21 topics were found in MEP and ES, respectively, where the main shared theme was the green supply chain. However, research on biofuels is unique to ES, and privatization is unique to MEP, among others. The concept of “performance” has also been covered by MEP and ES researchers. This article provides an objective snapshot of current research trends based on quantitative data, and the findings may be used to guide future research directions at the intersection of MEP and ES.

## 1. Introduction

Environmental sustainability (ES) is now a core concept embedded in several aspects of manufacturing enterprise operations. The shift toward greener production is driven by pressures from different fronts. From a market perspective, consumers are gravitating toward altruistic behavior, preferring eco-friendly despite higher prices; this behavior prevails if they are aware of the environmental benefits [1]. Within firms, changes to green practices may be perceived as costly and with no immediate rewards; however, managers also realize the value of sustainability as a key factor for innovation and introducing competitive advantage [2]. Environmentally sustainable practices also improve the social and financial performance of firms in the long term [3].

External pressures also play a role; government regulations are being established in developed and developing economies [4]. Additionally, international agendas such as the Sustainable Development Goals from the United Nations include specific entries and calls to action concerning clean production [5]. With these forces in play, researchers have devoted their time to understanding the implications of environmentally friendly practices for firm performance. As a result, there has been an ever-growing number of publications on sustainability- and manufacturing-related topics, making it difficult to keep track of trends.

Despite the growing publication trend, researchers have made sense of a large corpus of knowledge due to advances in data availability, computational power, and data mining methodologies. For instance, the number of publications on sustainability research has substantially increased, where a distinctive interdisciplinary field, sustainability science, has been created in which ES is a component [6]. Tracking the evolution of this field alongside the sub-topics that have arisen is possible due to data mining techniques; this is evident in bibliographic studies covering the taxonomy of sustainability [7,8]. This has also occurred for the research on manufacturing firms, which is expectedly larger given its maturity. While studies attempting to analyze the entire corpus on manufacturing enterprises appear scarce, those targeting specific sub-topics abound. For instance, Muhuri et al. [9] studied the literature of Industry 4.0 to identify sub-trends related to cyber-physical systems, the Internet of Things (IoT), smart industry and manufacturing, and industrial wireless networks. Similar studies have been conducted on smart manufacturing [10] and green manufacturing [11].

The overlap between sustainability and manufacturing has also been previously researched. For instance, Jian and Qu [12] visualized the network of articles related to sustainable manufacturing, uncovering four larger trends: additive manufacturing, power consumption, the green supply chain, and green information systems. Furstenau et al. [13] applied a similar approach to study articles on sustainability and Industry 4.0, revealing 12 trends based on keyword analysis. These articles have contributed to understanding current trends in these fields and research conducted at the intersection of these fields. However, it is still unclear as to how the specific field of ES relates to the performance of manufacturing firms. When researchers discuss firm performance, they may refer to financial, environmental, innovative, or other types of performance. The association between manufacturing enterprise performance (MEP) and ES research is unclear. More importantly, the gaps between the two fields that may be addressed by future studies are not known, particularly as both fields have grown independently, with sub-fields providing an opportunity for knowledge cross-sharing.

To date, there have been no publications that have conflated the research on MEP and ES; this study fills this gap. The objective of this study is to present an academic landscape of the MEP and ES literature to identify knowledge cross-sharing opportunities and gaps. This study also investigates how different types of “performance” are covered in both research corpora.

When overviewing academic domains, researchers opt for an expert-based approach or a data mining approach [14]. While overviews by experts are authoritative, they may be prone to bias toward their topics of interest or overlook related trends observed in other fields. Data mining approaches introduce comprehensive and accurate views of the fields, regardless of the size of the available literature. The application of data mining methods also enables the ability to reproduce results. These methods provide a quantitative snapshot that may be used for comparisons in future assessments of the fields. This study applied data mining methods to derive conclusions.

## 2. Materials and Methods

To surface common and distinctive topics between MEP and ES, a two-stage approach was followed. First, the MEP and ES research literature was analyzed separately. For both fields, a topical classification was identified based on the relationship among the academic articles published in each field. Second, the vocabulary of topics in MEP was compared to that in ES in order to establish a semantic connection or relatedness between the two fields. This approach assisted in highlighting common grounds and gaps that may represent opportunities for future synergies. Section 2.1, Section 2.2 and Section 2.3 provide the details of each stage in the method.

### 2.1. Stage 1: Citation Networks and Clustering

MEP and ES research was analyzed by following the steps shown in Figure 1. The academic literature on each topic was extracted from a bibliographic database. The Web of Science Core Collection was the selected source of data due to its broad coverage across the sciences, social sciences, arts, and humanities, where it has indexed over 21,100 peer-reviewed academic journals published worldwide. The database was queried by conducting a topical search: “TS = (“manufactur* enterpris*” OR “manufactur* industry*” OR “manufactur* firm*” OR “manufactur* company*”) AND TS = “perform*””. This search retrieved any article, conference proceeding, or book chapter relating to MEP. Similarly, a topical search: “TS = “environment* sustainab*””, was conducted to obtain articles relating to ES. Search strategies, including synonyms of the keywords (shown above), have been applied in the previous bibliometric literature mapping manufacturing- or sustainability-related topics. The search strategies used in these previous studies introduce off-topic clusters that are outside the scope of the present research. For instance, bibliometric studies analyzing the broad keyword, “sustainability,” also introduce sub-topics unrelated to the environment, along with generic and uncategorizable sub-topics [8]. To date, there have been no studies that have explored the landscape of “performance” in relation to manufacturing enterprises. As such, this study did not expand query terms or other synonyms; this was conducted to focus strictly on literature addressing these topics based on the exact queries in the title, abstract, or keywords of the articles. At the date of retrieval of 24 January 2021, there were 13,292 and 16,399 articles for MEP and ES, respectively. The search included articles published from 1900; this is the oldest year available in the database. The earliest publication of MEP occurred in 1973, while that for ES occurred in 1990.

Academic articles were then treated as nodes, and the connection between nodes was drawn when an article had cited or was cited by any other article in the dataset. To establish connections, a reference list provided by the database was used. References were found in the column, “Cited References” (CR), in the dataset. The document object identifier (DOI) in the reference list was matched to the articles in each dataset, which appears in the DI column in the dataset. When an article did not have a DOI, it was matched by comparing the author, year, publication source, volume, and issue number. This straightforward linkage is known as a direct citation network [15]. Although other citation-based networks exist (e.g., bibliographic coupling or co-citations), direct citations are proven to extract robust taxonomies of research fields [16] and are useful to identify academic fronts [17]. Some articles in the dataset were unconnected; these were articles that matched the queries during the retrieval step. However, they did not belong to the research topic due to the lack of connections to the main research corpus. As such, results were derived only from the giant component of the network. Lastly, articles were classified into topics by dividing the network into clusters.

Clusters refer to groups of intertwined nodes. An optimal clustering solution groups nodes such that there are more connections at the intra-cluster level than the inter-cluster level. This was evaluated by measuring the modularity of the network; the best partition is obtained when modularity is maximized [18,19]. In this study, the Louvain method was used for community detection [20] to obtain a clustering solution that maximizes modularity. This algorithm has been applied to citation networks in a variety of topics and is known to scale well in large networks [21]. Compared to other modularity maximization algorithms (e.g., Newman [22]) that tend to produce a mix of a few very large clusters followed by a large number of small clusters [23], the Louvain algorithm produces fewer clusters; this makes it easier to interpret the large network trends.

For both networks, the number of articles, the average year of publication, and the average citations received by articles in each cluster were computed. Clusters were named based on their most common keywords and the content of their most-cited articles. Networks and their clusters provide unbiased taxonomies of research fields as they are based on the “natural” citing behavior of researchers. Subsequently, the relationship between clusters in both networks was investigated.

### 2.2. Stage 2: Semantic Linkage

Clusters across networks were compared by examining vocabulary. Clusters sharing multiple keywords were considered to belong to the same or related topic, despite originating from different articles. To compute the similarity, this study followed a bag-of-words approach in which a vocabulary vector was created for each cluster. The vector was able to obtain the size of the total vocabulary in MEP and ES articles, extracted from the title, abstract, and keywords of articles, by concatenating, lowercasing, removing stop words, and obtaining the stem of each word. Each cluster vector reflects the count per keyword of the vocabulary found in articles of the cluster. These vectors were then weighted by the term frequency-inverse document frequency (tf-idf); this is a measure of the importance of a given keyword for a cluster in relation to the corpus [24]. Then, the cosine similarity score was computed using these vectors. Cosine similarity with tf-idf is a common and effective strategy to determine the similarity of documents in text mining [25]. While multiple similarity measures exist, vocabulary comparisons have been found to yield similar results to other metrics, particularly for technical vocabulary and in the social sciences [26]. Moreover, cosine similarity best approximates the consensus of domain experts when manually comparing cluster content [27]. In this study, semantic linkages refer to the similarity scores between pairs of clusters.

Figure 2 summarizes this stage, where clusters with a cosine similarity above average were interpreted as topics that existed in both networks. The remaining clusters contain vocabulary rarely observed in clusters of the other network, making them distinctive to MEP or ES.

### 2.3. Implementation

Data were acquired through direct download from the database website (https://apps.webofknowledge.com/ accessed on 24 January 2021) as tab-delimited (UTF-8) export files, including the full record and cited references. Data were processed in a Windows 10 personal computer with statistical software, where the R version 3.6.3 programming language was used [28]. The package igraph version 1.2.5 [29] was used to create the network and obtain clusters, and the package tm version 0.7.7 [30] was used for text processing. Nodes and edge files to replicate the network are provided in the Supplementary Material.

For visualization purposes, the plot of the network was established by applying a large graph layout (LGL) [31]. In this research, the selection of a layout algorithm over others did not impact the results or analysis of the networks; as such, the selection of LGL was based on computational efficiency. During the plotting of the network, only edges were shown which were assigned different colors to represent each cluster. In network visualization, clusters that appear in close proximity to other clusters are considered more topically similar.

## 3. Results

The giant connected component of the MEP research network consists of 7308 academic articles covering 55% of the articles in the dataset. The second largest component contained only 13 articles (0.1% of the dataset). ES research shows a similar pattern with 6275 articles within its giant component covering 38% of the dataset, followed by a component of size 20 (0.1% of the dataset). As such, the following results focus on the giant components as they contain the core of the research in each network. The remaining articles in the dataset were unconnected and did not form clusters that had a sufficient volume of publications to be considered. Figure 3 shows the number of publications per year, which was characterized by an increasing trend. In 2016, more academic articles relating to MEP were published, although the trend subsequently shifted with a decline in MEP research in the past year. MEP is also a more mature topic, with the earliest papers in the dataset published in 1973; these papers tackled the economic performance of the manufacturing industry in relation to diversification [32], and market structure [33]. Conversely, the earliest articles on ES were the works of Barbier et al. in 1990 on cost–benefit analysis for environmentally sustainable development [34,35]. Since then, the academic landscapes of MEP and ES have grown and changed, covering a variety of topics.

### 3.1. The Academic Landscape of MEP

Figure 4 presents the academic landscape of MEP. Academic articles were classified into 16 different clusters, with 341 residual articles classified as “other,” for a total of 17 clusters. From the network, the topics of innovation and lean manufacturing were central to MEP, while the remaining articles were more distributed. Table 1 shows the key statistics of each cluster, followed by a description of notable clusters.

The largest cluster in MEP was on innovation; this topic covers research on the role of open innovation in the innovative performance of firms [36], and of collaboration in new product development [37], among other aspects of the innovative process. In terms of the number of articles, this cluster was followed by another two related to the supply chain.

The supply chain management literature explores the impact of different supply chain management practices to enhance competitive advantage and organizational performance [38]. This cluster focuses on theories and case studies for management practices of the supply chain that are within and beyond the factory [39].

The green supply chain refers to the adoption of environmentally sustainable practices to mitigate and control pollution, spanning from the production process, through delivery, to the user. This cluster collects research on the driving and enabling factors that contribute to a green supply chain [40], and studies on the relationship between green supply chain practices and the economic performance of firms [41,42]. This is also the newest cluster in terms of the average publication year of its articles.

The smallest cluster was on privatization; this topic discusses the effects of transferring publicly owned manufacturing companies to private capitals, and its impact on the economic performance of firms and impacts on the government [43]. This cluster also explores how privatization affects the intellectual capital of firms [44].

The oldest clusters were enterprise resource planning and production planning, both of which are topically related. The former covers research on implementation and investment in information technologies to improve firm performance [45,46]. Production planning focuses more generally on operations research than the information system used by the firm [47].

Energy efficiency was the second youngest cluster and covers studies on strategies for energy saving and emissions reduction in manufacturing companies [48]. It includes research on management tools, indicators, certification, and other mechanisms to measure and validate energy efficiency during production [49].

Table 1 shows the average citations received by articles in the clusters; articles receive citations for academic and non-academic reasons such as supporting claims or discussing disagreement [50]. In general, associations between higher citations with better academic performance have been disputed [51]. In this article, the average citations of clusters were considered a signal of academic interest: positive or otherwise. The cluster with the highest average citations was that for market orientation. This research focuses on studying the performance of manufacturing firms adopting a philosophy centered on developing products that satisfy user needs, as opposed to a product-oriented philosophy in which products are “forced” into the market through marketing and delivery strategies [52]. In looking for a market-oriented approach, firms adopt a corporate entrepreneurship mindset [53]. The cluster with the lowest average citation was the optimization system and algorithm. This cluster collects research on the computational and mathematical models for energy efficiency [54], and production scheduling [55].

### 3.2. The Academic Landscape of ES

Figure 5 presents the network representing the ES academic landscape. There were 20 main clusters and a residual cluster; the ES network was sparser than that of MEP research. The cluster of sustainability indicators was central to the network, although the cluster on green supply chains was dominant based on the number of academic articles and its more dense connections compared to other clusters. Table 2 summarizes the bibliometric statistics for each cluster in ES.

The green supply chain was the largest cluster; this topic appears in the MEP network with research focused on the manufacturing firm perspective. In ES, the scope of this cluster was much broader. Research in this cluster focuses on logistics service providers [56], the service industry, and other stakeholders [57,58] in their role to achieve a sustainable supply chain.

The next largest cluster was that of sustainability indicators. This cluster covers research on developing measurement and evaluation frameworks for indicators related to sustainability [59]. A variety of indicators currently exist, including the ecological footprint, the Environmental Sustainability Index, and the Dashboard of Sustainability, while other indicators are often compared and reviewed to achieve incremental improvements [60].

The smallest cluster pertains to green buildings and sustainable construction. This cluster includes discussions on practical changes to help traditional building practices achieve cost performance in green building projects [61]. It also covers the development of certification, standards, and indicators to evaluate green construction projects [62].

Trade policy and economic growth were clusters that contained publications with the youngest average publication year. This cluster covers research on the co-dependence on energy among countries and regional clusters [63], and how different trade and monetary policies affect the carbon emissions of a given country [64].

The agrifood supply chain cluster contained the oldest average publication year. Studies in this cluster include analyses on how the influence of societal change on responsible consumption has favored shorter supply chains, with a preference for local products [65]. It also included studies on how technologies such as the IoT positively affect the environmental performance of supply chains in agriculture [66].

Household sustainability research was the second oldest cluster, while also being the cluster that had more citations, on average. This cluster covers the theoretical discussion and conceptualization of sustainable development, with an emphasis on sustainable livelihood and well-being [67]. It also includes research exploring the concept of telecoupling, which refers to how living, working, or retiring in remote locations impacts ideas of sustainability [68].

Sustainable universities was the cluster with the lowest average citations. This clusters includes studies on how sustainability is being or should be included in the curricula of schools and universities [69,70]. It also covers studies on how to create sustainable campuses [71].

The cluster on biofuels covers research on the environmental impacts that occur during biofuel production. For instance, the advantage of reducing greenhouse gas emissions compared to fossil fuels appears clear, although when compared to other sources of energy, this impact remains uncertain [72]. However, multiple studies have attempted to identify the best bio-refinery process and mechanisms to mitigate such impacts [73] or explore the types of biofuels that are better under different conditions [74].

### 3.3. The Linkage between MEP and ES

The commonalities and differences of MEP and ES were uncovered by computing the similarity between pairs of clusters based on shared vocabularies; Figure 6 illustrates this linkage. In the figure, clusters are sorted from top to bottom; those with a higher cumulative semantic similarity to clusters in the opposite network were placed on the top, and those with the lowest similarity were placed at the bottom. Using this sorting, it was possible to navigate both academic landscapes and identify research opportunities across topics that were beyond those already apparent or saturated, such as green supply chains, energy management, and Industry 4.0. While research on these topics is still necessary and encouraged, other topical research gaps remain poorly studied.

The highest similarity (0.765) was noted between ES cluster 1 (ES-1) and MEP cluster 3 (MEP-3), relating to the green supply chain. This was anticipated as this topic appears in both landscapes, making it a pivotal topic. The clusters on supply chain management (MEP-2), lean manufacturing (MEP-4), and innovation (MEP-1) in relation to the green supply chain (ES-1) were the following three pairs with high similarity scores. The cluster on green supply chains in the ES network covered a variety of research sharing the specific topics and keywords of these three clusters. While research on supply chains is central, there are also discussions on the implications of lean quality and operation methods such as lean and Six Sigma for improving environmental performance [75], and the role of logistics service providers in green innovation [76].

The next most similar clusters were energy efficiency (MEP-7) and Industry 4.0 (ES-12). These clusters were expected to observe discussions related to energy management in the context of a sustainable Industry 4.0. The similarity of these clusters may be attributed to research on minimum quantum lubrication (MQL); this is a technique that facilitates near-dry machination by injecting a reduced amount of oil through compressed air. While MQL has been researched in the context of improving the production performance of manufacturing companies, it has also been discussed as a means to achieve sustainable production [77].

Distinctive clusters have low or no similarity to clusters in other networks. By checking the cumulative similarity scores of the clusters, the clusters of privatization (MEP-16) and production planning (MEP-14) were the most distinct of MEP. The clusters of biofuels (ES-5) and health and food consumption (ES-3) were the most distinct of ES.

## 4. Discussion

MEP and ES are two independent fields of research that have recently begun to overlap from a bibliographic perspective. A total of 108 academic articles were identified in both networks; this means both of them contained keywords from the search strategy and thus focused on MEP and ES simultaneously. As a group, these articles have an average publication year of 2016.8, and a median of 2018; this means that half of the articles have been published in the past three years. Despite the presence of these intersecting articles, research on MEP and ES remains largely disjointed. The intersecting articles accounted for only 1.5% of MEP articles and 1.7% of the ES literature; 55 of these articles were focused on the green supply chain. The full list of articles, including their placement in the clusters, is provided in the Appendix A.

Green supply chain management is the main transversal topic across the MEP and ES fields. Although this topic has been studied with some nuances, the common ground shared between articles in clusters is that of an observed benefit on economic and environmental performance when green supply chain management is well executed. In this regard, knowledge of well-established and classic operational principles, such as quality management and just-in-time, may be transferred from the production plant to the entire supply chain to achieve environmental goals [41]. The variety of agents studied in the clusters of the green supply chain highlights the need for orchestrated efforts at the business ecosystem level. At this level, the responsibility of better environmental practices in manufacturing does not rely solely on the manufacturer; rather, this is also distributed to the government and other institutions. Environmental collaboration with suppliers and customers and the role of standardization and certification organizations were observed as key drivers of environmentally sustainable supply chains [40].

Researchers working at the intersection of manufacturing and sustainability are likely to recognize that green supply chain research has played a substantial role in improving the economic and environmental performance of firms [78]. For instance, green practices within the factory related to waste reduction [79] or imposed through regulation (e.g., CO_2_ emissions control) [80] serve as examples of topics that have matured. This article provides quantitative evidence of this overlap, based on the presence of intersecting academic articles and through investigating the shared vocabularies used by researchers when studying multiple topics within the bounds of MEP and ES. Opportunities for future research have also been identified by observing the differences between the two landscapes.

For instance, although biofuels is the fifth ES cluster in terms of the number of publications, there were no articles identified at the intersection of MEP. An additional search revealed only 12 articles when searching biofuels and manufacturing firms in the Web of Science (i.e., not related to “performance”); these were mostly in relation to the manufacturing of biofuels without addressing environmental implications [81]. Similarly, little research has been conducted on the privatization of manufacturing firms and its implications for ES. Another perspective to identify knowledge gaps is to observe relatively new trends that have less similarity to their counterparts. In this manner, MEP scholars may include sustainability into the scope of ongoing studies associated with optimization, algorithms systems, and quality standards and certification. However, little research has explored the benefits or case studies on both. ES scholars may explore manufacturing performance in the context of trade policies, economic growth, and food consumption. These two trends have recently begun expanding in ES research, although their connection to manufacturing research is lacking.

Finally, the usage of the term “performance” is discussed. This article reveals the academic landscape of MEP, where researchers refer to different types of firm performance. From the title and abstracts of MEP articles, all instances of the term “performance” were extracted along with any other adjacent term. Table 3 lists these instances mentioned in more than 100 papers. Aside from the generic instance of “firm performance,” it was observed that MEP researchers were mainly interested in innovation and financial performance, based on the number of publications; the environmental performance of firms followed after these topics. On the other hand, ES scholars were more interested in economic and financial performance, in addition to environmental and sustainability performance.

Figure 1 demonstrates that the concentration of articles on ES has only occurred recently; this is also true for research on environmental and sustainability performance. In both cases, there was little variation in terms of the publication year of related research when compared to their publication in MEP or ES fields.

Interestingly, articles on operations, quality, and manufacturing performance published as ES research received a higher number of citations on average, despite being so small in number. Articles on environmental performance received mid-range citations, and sustainability performance was the type of performance in which the related articles received the lowest citations on MEP.

Based on these findings, it may be argued that there is an unbalanced treatment in the study of firm performance between these two fields of research. MEP researchers may have begun publishing more papers on environmental and sustainability performance, although such articles are not cited as much as those focused on innovation and economic performance. On the other hand, when ES scholars publish articles related to manufacturing, such as quality, process, and operations performance, they tend to attract greater citations from the overall academic community. This may be interpreted as a call to action to establish synergies between the two fields; it signals the positive outcomes that may be possible with an interdisciplinary effort.

Investigating the academic landscapes presented in this paper may potentially contribute to the advancement in the MEP and ES fields. In the context of research evaluation, funding, and policy making, this paper presents state-of-the-art research on MEP, ES, and its commonalities. Evaluators and funders of new research proposals are provided with a quantitative and objective perspective on topics already well covered, emerging topics, and topics that are under-researched. Visualizing and classifying fields of research in this manner support evidence-based decision making, as they enable an assessment, to some extent, on future avenues for research that are original or necessary. Experts and practitioners are also able to target new topics for synergies. Researchers seeking to advance their fields are able to position themselves in a determined cluster and identify topics in other research fields that may represent better academic collaborations. Those that are new to research are also provided with an overview of the MEP and ES fields; using this overview, they are able to begin exploring gaps and brainstorming ideas for new research.

Although the bibliometric methods applied in this research have been widely used for academic landscaping and literature discovery, they also present some limitations. First, in terms of data retrieval, specific search queries were used to target research on manufacturing performance and ES. It may be argued that other queries may have potentially introduced greater or fewer articles, or that the concept of “sustainability” alone implies “environmental sustainability”. Previous research has shown that sustainability is a concept that covers multiple aspects, and it is not only limited to the environment [8]. As such, defining “environmental sustainability” from an information retrieval perspective remains an open question. Another limitation lies in the nature of direct citation networks, where the cluster structure changes as articles obtain a greater number of citations over time. As such, other clustering approaches that are time-independent are worth exploring. For instance, clusters are based on topic modeling; however, such text-based methods are also disputed due to instability and difficulty with reproducibility [82]. The final limitation is associated with the analysis of gaps. Although distinctive topics of MEP and ES were identified, it is not clear whether any possible combination of cluster pairs deserves further research. Efforts to create frameworks or methodologies to assess the relevance and urgency of addressing determined topical gaps present a host of opportunities in future research.

## 5. Summary and Conclusions

This article explored the academic landscapes of MEP and ES by applying an unsupervised classification system that takes advantage of the citation networks of articles published in these fields. The findings showed that MEP could be divided into 17 topics, while ES was divided into 21 topics. A semantic linkage was then conducted to identify commonalities and differences between the two networks. The pivotal topic shared between the two fields was the green supply chain, where 788 and 859 articles were published in MEP and ES, respectively. Other topics with high similarity between networks were related to energy efficiency and Industry 4.0. There were also topics that were distinctive for each field, such as biofuels in ES and privatization in MEP. Moreover, it was found that 20 types of “performances” were discussed in very different ways in both fields. Although innovation and financial performance were the focus of MEP researchers, environmental performance followed based on the number of publications on this topic. ES scholars were less interested in research related to firm performance; however, when a paper on this topic was published, these papers tended to be highly cited. The exploration of landscapes presented in this article provides quantitative evidence on the commonalities that are well studied, emerging trends, and gaps that may represent a green field for future research.

## Figures and Tables

**Figure 1 ijerph-18-03370-f001:**
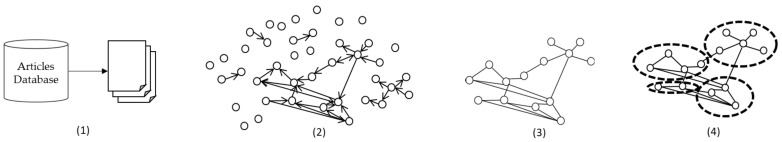
Method to create a direct citation network. (**1**) Data from extracted articles including reference list from the database; (**2**) establish a connection among articles based on citations; (**3**) extract the giant component; (**4**) clustering.

**Figure 2 ijerph-18-03370-f002:**
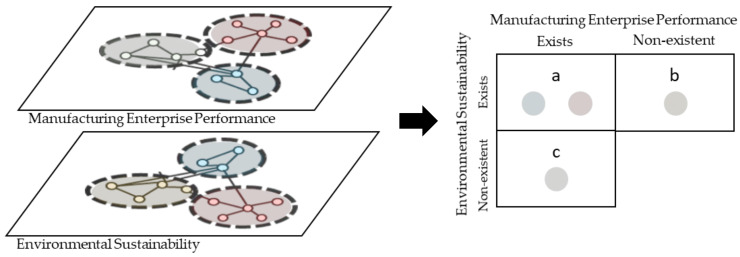
Identifying linkages between two topics of interest. Clusters sharing similar semantic content signal the presence of shared topics (quadrant “a”), while the remaining clusters are distinctive topics (quadrants “b” and “c”).

**Figure 3 ijerph-18-03370-f003:**
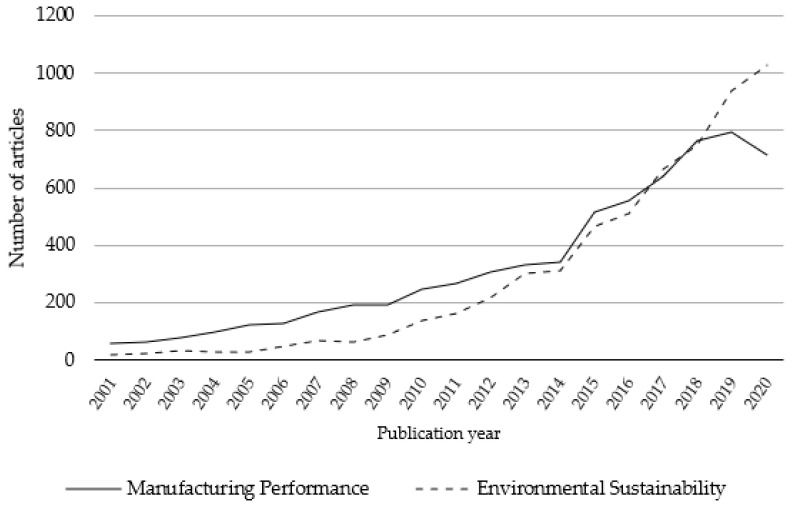
Publication trends for manufacturing enterprise performance (MEP) and environmental sustainability (ES) over the past 20 years.

**Figure 4 ijerph-18-03370-f004:**
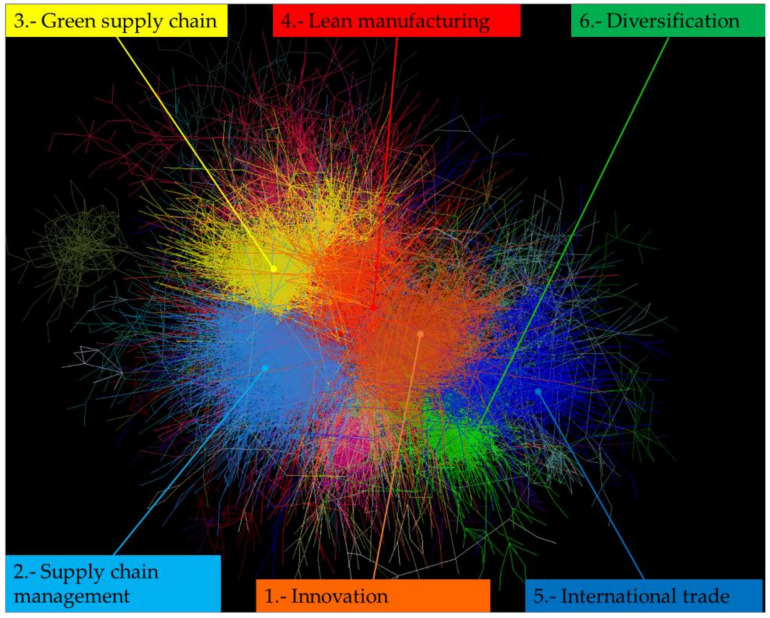
Citation network of manufacturing enterprise performance research. Each colored section represents one of the 17 clusters that were identified; the six largest clusters are labeled.

**Figure 5 ijerph-18-03370-f005:**
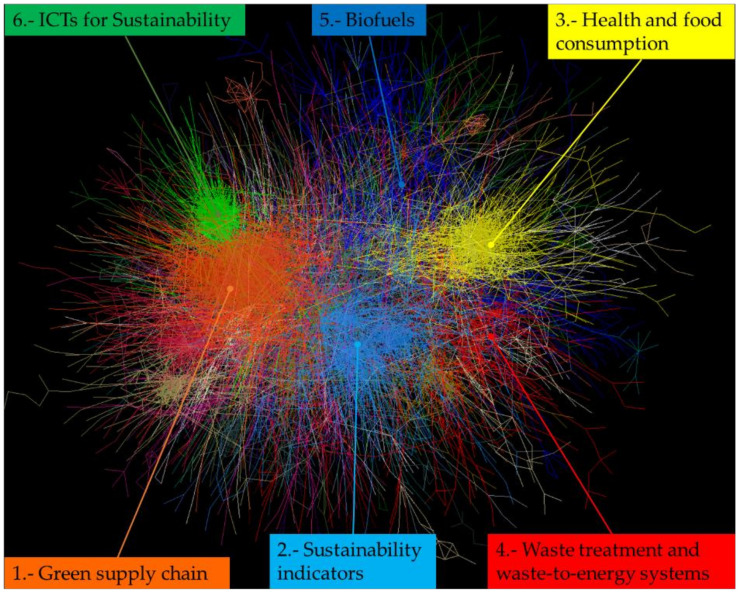
Citation network of environmental sustainability research. Each colored section represents one of the 21 clusters found; the six largest clusters are labeled.

**Figure 6 ijerph-18-03370-f006:**
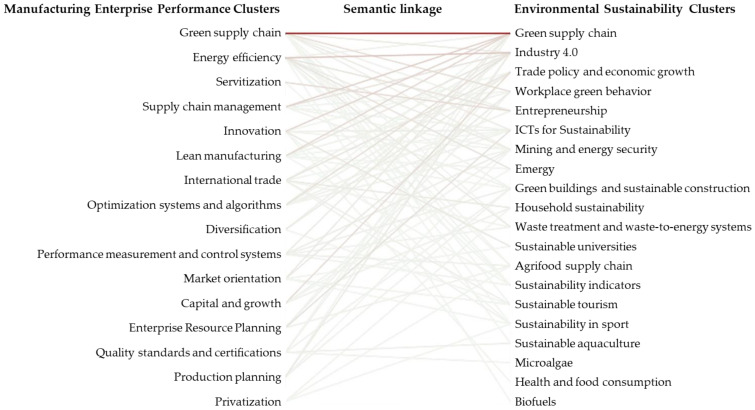
Semantic linkage between MEP and ES. Clusters were sorted from the one most related to clusters in the other network. Those at the bottom are the most distinct clusters.

**Table 1 ijerph-18-03370-t001:** Clusters for manufacturing enterprise performance.

Cluster	Cluster Name	Articles	Average Year *	Average Citations *
1	Innovation	1176	2015.0	37.4
2	Supply chain management	1139	2013.2	30.6
3	Green supply chain	788	2016.3	31.3
4	Lean manufacturing	676	2013.4	22.5
5	International trade	630	2012.7	25.9
6	Diversification	407	2011.6	45.5
7	Energy efficiency	370	2016.1	23.9
8	Capital and growth	295	2011.1	22.7
9	Market orientation	274	2011.7	54.9
10	Performance measurement and control systems	268	2011.6	29.2
11	Servitization	227	2015.6	29.7
12	Enterprise resource planning	173	2006.8	37.3
13	Quality standards and certifications	150	2014.8	22.1
14	Production planning	144	2009.8	23.3
15	Optimization systems and algorithms	143	2015.7	18.1
16	Privatization	107	2012.2	21.5
17	Other	341	2012.5	18.7

* Summary statistics including minimum, 1st quartile, median, 3rd quartile, and maximum values are included in the Appendix A.

**Table 2 ijerph-18-03370-t002:** Clusters of environmental sustainability research.

Cluster	Cluster Name	Articles	Average Year *	Average Citations *
1	Green supply chain	859	2016.5	29.9
2	Sustainability indicators	564	2015.0	22.9
3	Health and food consumption	454	2016.9	29.5
4	Waste treatment and waste-to-energy systems	331	2016.3	27.5
5	Biofuels	324	2016.2	33.8
6	ICTs ** for sustainability	270	2015.3	19.3
7	Workplace green behavior	252	2016.3	26.8
8	Agrifood supply chain	223	2014.3	30.3
9	Trade policy and economic growth	210	2017.8	18.9
10	Sustainable tourism	198	2015.7	23.4
11	Mining and energy security	184	2015.6	20.0
12	Industry 4.0	179	2016.7	22.0
13	Sustainable aquaculture	172	2015.5	34.8
14	Household sustainability	169	2014.5	75.0
15	Sustainability in sport	160	2015.1	20.3
16	Microalgae	160	2015.5	52.2
17	Entrepreneurship	159	2016.3	24.5
18	Sustainable universities	146	2016.1	17.4
19	Emergy	134	2014.6	22.7
20	Green buildings and sustainable construction	132	2015.2	33.3
21	Other	995	2015.9	23.0

* Summary statistics including minimum, 1st quartile, median, 3rd quartile, and maximum values are included in the Appendix A. ** Information and communication technologies.

**Table 3 ijerph-18-03370-t003:** Instances of “performance” keywords in the MEP and ES literature.

Keyword	MEP			ES *		
	Articles	Average Year	Average Citations	Articles	Average Year	Average Citations
Firm performance	1532	2013.6	40.9	103	2016.5	36.2
Innovation performance	1215	2015.2	41.3	21	2018.2	6.2
Financial performance	882	2014	36.3	103	2017.1	26.9
Environmental performance	607	2015.8	33.5	678	2016.3	28.4
Business performance	525	2013.8	28.9	43	2017.5	13.5
Operations performance	488	2014	24.3	45	2016.2	57.5
Organizational performance	356	2012	37	42	2017.5	31.5
Sustainability performance	302	2017.4	16.2	330	2017.4	19
Export performance	268	2011	28.6	2	2017	16
Economic performance	266	2012.2	42.8	116	2015.9	40
Product performance	239	2012.5	27.4	13	2016	24.9
Supply chain performance	228	2014.7	26.3	28	2017.6	22.7
Manufacturing performance	224	2012.6	27	6	2014	376.3
Improve performance	202	2013	41.8	26	2016	55.3
Process performance	171	2011.2	19.9	14	2018	41.2
Company performance	144	2012.8	34.3	8	2018.6	6.4
Market performance	144	2012.2	32.7	18	2018.1	26.2
Quality performance	123	2010.8	40	4	2014.5	90.3
Organization performance	116	2013.2	27.5	6	2020	3.5
Supplier performance	102	2013.1	48.3	10	2016.5	64.3

* Relations of keywords to ES clusters are provided in the Appendix A.

## Data Availability

Raw data for network reproducibility are included in the Supplementary Material. A full bibliographic dataset is available for those with a Clarivate Analytics’ Web of Science Core Collection license by querying the data as instructed in the Methods section. Other data are available upon request to: mejia.c.aa@m.titech.ac.jp.

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
