# Peer review of "The Academic Landscapes of Manufacturing Enterprise Performance and Environmental Sustainability: A Study of Commonalities and Differences"

_ijerph, 2021, doi:10.3390/ijerph18073370_

Round 1

Reviewer 1 Report

The paper is prepared correctly. It includes all necessary chapters, quality and visibility of figures are sufficient. I don't have any comments to the edditting part. From the other point  of view, substantively the content of the article brings an average novelty. There is a little lack of investigations, results or modelling. Used methods are advanced but materials are only a bibliometric indexes. In my opinion maybe the "Review type" of paper would suit more? Summarized - the paper could be published  with an averege novelty but high quality of presentation.

Reviewer 2 Report

In this paper, a literature of MEP (manufacturing enterprise performance), and ES (environmental sustainability) was overviewed in order to find their common and distinctive topics. A topical classification based on the relationship among the published academic articles was carried out on MEP and ES separately. Afterwards, authors compared the vocabulary of the MEP and ES topics in order to establish a semantic connection or relatedness between the two literatures.

It constitutes a very interesting topic deserving in depth-study and publication in the scientific literature. However, this reviewer has many concerns with regard this study due to it is based on specific search queries; the selected literature articles are heavily dependent on them! The results and the considerations in the manuscript are related only to the selected articles. This makes the study an end in itself. This limit was also raised by the authors (lines 371-383). This reviewer would like a convincing answer with regard to this aspect in order to accept the document.

Moreover, from this reviewer's point of view, this paper needs a considerable work to be accepted for the publication:

  • Introduction is very concise and simple;
  • Please remove the personal form (WE, WE, WE…..) from throughout the manuscript;
  • Figures and Tables should be mentioned in the text wherever appropriate;
  • specify what is l in equation 1;
  • increase the resolution of the Figure 3;
  • table 2 should not be divided in two pages;
  • The poor quality of the English used makes the text often difficult to understand and to proofread. A substantial improvement by an English native speaker is advised.

Reviewer 3 Report

First, the authors use the "off-the-shelf" analytical tools, which is good, but in the text, they are unnecessarily (dis)focusing the readers on the selected theoretical formulas. It would be enough to refer readers to appropriate literature than to present an individual formula (1)(2) that is not covering in complete issues being touched anyway. Those familiar with SNA, and NLP need not see these maths once again, and for those who are not, it would be not enough. Instead, the authors could describe the toolchain being used during this research. It is in particular important in the current era of research repeatability, and would also be helpful for those wanting to improve their scientific workshop. So the process and methods of data acquisition (API vs file downloading), data cleaning (manual vs computer-aided), and data analysis (R, Python, Mathlab, libraries used...) with respect to tools being used for conducting presented research should be specified.
The second general issue is the very synthetic presentation of results. At the beginning of the paper, the authors write: "While overviews by experts are authoritative, they may be prone to bias towards their topics of interest, or to overlook related trends observed in other fields", and I agree basically. However, experts would form more practical/informative conclusions, than only showing the number of citations. Thus, I strongly encourage the authors to go a little bit further and to try to explain/deduce what results from presented numbers.
Finally, I ask authors for uploading here also raw data (also that allowing for drawing network structures).

Detailed tips follow.
[58] 'literatures' - I doubt if this plural form is being used commonly
[62] please be more specific - this is not a computer approach - this is a quantitative approach, computers are only helpers here
[79] broken reference (further BR)
[88] the passive voice would be more appropriate
[90] so we know the end of the analyzed time window, what is the start?
[96] be more specific - please describe what data attribute allowed for implementing these "connections" - was it some technical ID, or you tried to automatically pair by analyzing the whole bibliographic records - I am familiar with WoS, but readers may not necessarily
[102] - sentence should start with CAP
[104] - consider using here also the term more commonly used - "giant component"; please tell readers what this giant component was a part of the whole graph by percentage, and what was the volume of the second biggest component to convince them that it doesn't make sense to analyze the rest
[105] - to cluster the directed graph in this way is sometimes disputable especially when this direction comes from timeline - the longer a given cluster lives the more sparse it is getting to be; but I accept this approach, please only explain that this is a directed network, and describe my doubts in the research limitation section
[106] - highly is exaggerating term here - it simply needs to be higher than others
[109] this is not an article on network clustering - please consider replacing this formula with reference to a good manual/review article
[112] please explain to readers why authors use this specific method - there are many other community detection algorithms
[114] please explain to readers why authors use this specific method - it is quite outdated and gives no spectacular result - the only advantage is low computational complexity, but, in my opinion, your network structure is not as big that this choice would be justified
[134] see [109]
[139] I argue there exists a more appropriate reference, that would also help authors to avoid self-citation
[142] BR
[153] BR
[166] BR
[170] BR
[190] the avg year is very misleading here, there may exist a hot topic being exploited from 2000 to 2020 intensely, and this measure would give the same result for the side topic that appeared and disappeared in 2010 - can you propose something else/additional to avoid such misinterpretations? also explain/propose to the readers how to interpret avg. citation measure in more general - to not be just numbers
[214] BR
[219] BR
[264] BR
[310] BR
[325] Fig.7 presents exactly the same as what the heatmap shows, so one should be removed - I would suggest removing the heatmap because it is harder to read. Then the content describing Fig.7 could be moved to Result section.
[332] BR
[339] BR

Round 2

Reviewer 2 Report

All points suggested by this reviewer are taken into account by the authors also with detailed answers. 

The personal form is still present in some parts of the text. Please check and remove it.

Author Response

Thanks for your revisions. The latest manuscript has been proofread and edited by a native, and the personal form removed.

Reviewer 3 Report

The recommended improvements were implemeted in a satisfactory manner. Now I recommended this paper for publication.

Author Response

Thanks for your revisions. Additionally, the latest manuscript has been proofread and edited by a native.